# Insights into the Synthesis Parameters Effects on the Structural, Morphological, and Magnetic Properties of Copper Oxide Nanoparticles

**DOI:** 10.3390/ma16093426

**Published:** 2023-04-27

**Authors:** Fatma Mbarek, Ichraf Chérif, Amira Chérif, José María Alonso, Irene Morales, Patricia de la Presa, Salah Ammar

**Affiliations:** 1Electrochemistry, Materials and Environment Research Unit, UREME (UR17ES45), Faculty of Sciences of Gabes, University of Gabes, Gabes 6072, Tunisia; 2Higher Institute of Education and Continuous Training of Tunis, Virtual University of Tunis, Tunis 1073, Tunisia; 3University of Tunis El Manar, Faculty of Sciences of Tunis, Laboratory of Materials Organization and Properties, Tunis 2092, Tunisia; 4Institute of Applied Magnetism, Complutense University of Madrid, A6 22,500 Km, 28230 Las Rozas, Spainpmpresa@ucm.es (P.d.l.P.); 5Institute of Materials Sciences of Madrid, CSIC, Sor Juana Ines de la Cruz, 28049 Madrid, Spain; 6Department of Material Physics, Complutense University of Madrid, Plaza de la Ciencia 1, 28040 Madrid, Spain

**Keywords:** CuO nanoparticles, oxalate precursor route, X-ray powder diffraction, TEM and SEM observations, magnetic properties

## Abstract

The present study aims at the integration of the “oxalic conversion” route into “green chemistry” for the synthesis of copper oxide nanoparticles (CuO-NPs) with controllable structural, morphological, and magnetic properties. Two oxalate-containing precursors (H_2_C_2_O_4_.2H_2_O and (NH_4_)_2_C_2_O_4_.H_2_O) and different volume ratios of a mixed water/glycerol solvent were tested. First, the copper oxalates were synthesized and then subjected to thermal decomposition in air at 400 °C to produce the CuO powders. The purity of the samples was confirmed by X-ray powder diffraction (XRPD), and the crystallite sizes were calculated using the Scherrer method. The transmission electron microscopy (TEM) images revealed oval-shaped CuO-NPs, and the scanning electron microscopy (SEM) showed that morphological features of copper oxalate precursors and their corresponding oxides were affected by the glycerol (*V*/*V*) ratio as well as the type of C_2_O_4_^2−^ starting material. The magnetic properties of CuO-NPs were determined by measuring the temperature-dependent magnetization and the hysteresis curves at 5 and 300 K. The obtained results indicate the simultaneous coexistence of dominant antiferromagnetic and weak ferromagnetic behavior.

## 1. Introduction

Nanostructured transition metal oxides, in particular MO-type metal oxides (M = +II transition metal ions: Cu, Zn, Fe, Mn, Ni, Co), are essential for the conception of various novel functional and smart materials [1]. Considerable efforts have been devoted to their preparation; however, synthesizing highly pure nanoparticles with a limited number of steps and eco-friendly low-cost methods to facilitate their industrial-scale implementation remains challenging. In recent years, the oxalate route, which is classified as a wet chemical method, has attracted much attention for synthesizing MO-type nanoparticles. This method is based on the precipitation of metallic oxalates MC_2_O_4_.nH_2_O followed by their thermal decomposition to the oxide stage in an appropriate atmosphere [2].

It is worth noting that among the above-mentioned metallic oxalates for which n = 2 and the coordination sphere of M^2+^ ions is octahedral, defined by two chelating oxalate ions and two water molecules, copper oxalate is particularly different. It has various hydrated forms presenting “zeolitic water” that does not affect the structure integrity [3]. The Cu^2+^ environment, although octahedral, is ensured by two chelating oxalate ions from the CuC_2_O_4_Cu ribbons and two oxygen atoms from oxalate groups belonging to other ribbons, as suggested by Fichtner-Schmittler [4]. The water molecules’ degree “n” seems to depend entirely on the precursor’s nature as well as the fabrication procedures. For instance, simple precipitation methods using copper nitrate and oxalic acid or copper sulfate and potassium oxalate as starting materials lead to the formation of anhydrous CuC_2_O_4_ and CuC_2_O_4_.0.5H_2_O, respectively [5,6]. Furthermore, Wenpei et al. [7] employed hydrothermal and solvothermal methods that led to the successful synthesis of n = 0.14 and n = 0.53 compounds. 

In addition to copper oxalate, copper oxide has also long been a special material owing to (i) its unique structural features and (ii) its original physico-chemical properties [8,9]. (i) With regard to the structural features, it must be noted that MO oxides crystallize in highly symmetrical crystal systems, notably cubic (FeO, MnO, CoO, NiO) or hexagonal (ZnO), which is not the case for CuO. Its structure was elucidated in the monoclinic system (space group: C 2/c) by Tunell et al. [10] in 1935. Both Cu^2+^ and O^2−^ ions have a coordination number of four, which defines different polyhedrons consisting of a distorted square plane (CuO_4_-SP) and a tetrahedron (OCu_4_-T), respectively. (ii) As far as the physico-chemical properties are concerned, it should be stated that CuO is the only binary compound known to be “multiferroic” [11], for which it is first necessary to understand each component separately for better monitoring and to achieve the “magnetic and ferroelectric” coupling. Furthermore, the magnetic properties of CuO still intrigue scientists due to the unexpected phenomena and surprising behaviors observed as functions of synthesis methods and operating conditions. CuO is especially different from the other magnetic metal oxides and its magnetism is perhaps the least understood among them [12]. Hu et al. (1953) [13] demonstrated that CuO in bulk shows a specific heat anomaly at 220 K associated with the onset of antiferromagnetism. More recently, Ota and Gmelin [14] determined an incommensurate-to-commensurate antiferromagnetic transition at a temperature of 213 K in a single CuO crystal by means of specific-heat studies. However, its magnetic susceptibility does not show typical Néel temperature behavior, but rather a constant susceptibility up to 130 K followed by a smooth increase up to 400 K. In addition, within the frame of this intriguing scenario, the magnetic behavior changes and deviates significantly from that of bulk material as the particle size decreases. As reported by Punnoose et al. [15], the Néel temperature decreases with the particle size. For those particles with sizes above 30 nm, the Néel temperature is close to that of the bulk and the thermal curve behavior resembles that of the bulk except at temperatures below 100 K.

On the other hand, it should be noted that the experimental synthesis conditions play a crucial role in tuning the morphological features of nanoparticles and consequently their physico-chemical properties. The reaction medium is one of the most important parameters that affect the crystallinity, shape, and the size of NPs by acting on the nucleation, growth, and aggregation/agglomeration kinetics [16]. Wu et al. [17], for instance, studied the effect of N,N-Dimethylacetamide (DMAC)/water mixed solvent on CuO morphology. Different volume ratios of DMAC and water were used, namely 1:5, 1:1, 2:1, 5:1, 7:1, and 11:1, among which (i) the 5:1 ratio led to the smallest CuO size of 6.7 nm and (ii) the highest DMAC amounts of 7:1 and 11:1 induced impurities via incomplete reaction. More recently, Zaid et al. [18] synthesized CuO-NPs through the irradiation of copper oxalate in different solvents (water, methanol, and ethanol) and found that as the dielectric constant of the medium decreased, the particle size decreased reaching 8.4 nm in ethanol compared with 11.4 nm in water. Similarly, Siddiqui et al. [19] demonstrated that solvent properties influence the crystallinity and the shape of CuO-NPs. They used ethanol and isopropyl alcohol as solvents in the sol-gel method and observed that isopropyl alcohol led to flake-like CuO-NPs with the better crystallinity compared with the non-uniform samples synthesized in ethanol. In the light of these examples and taking into account the environmental considerations that recommend the use of ecofriendly solvents, glycerol (C_3_H_8_O_3_) seems to be a potential candidate that can be introduced as a “greener solvent” or “organic water” for synthetic chemistry [20]. Glycerol is a natural polyol generated by the vegetable-oil industry, a polar protic solvent completely soluble in water, that is non-volatile under atmospheric pressure, non-toxic, biodegradable, non-flammable, inexpensive, and available on a large scale [21,22,23].

Thus, based on the above-mentioned facts, the present study aims to investigate:

the combination of the so-called “oxalic conversion” or “oxalate precipitation” route with green chemistry by using glycerol as a green solvent to prepare CuO-NPs. the impact of different proportions of glycerol and oxalate ion precursors (oxalic acid and ammonium oxalate) on the structural, morphological, and magnetic properties of the green synthesized CuO-NPs. 

## 2. Materials and Methods 

### 2.1. Materials 

All reagents were purchased from Sigma-Aldrich and directly used without further purification. Anhydrous copper (II) chloride CuCl_2_, oxalic acid dihydrate H_2_C_2_O_4_.2H_2_O and ammonium oxalate monohydrate (NH_4_)_2_C_2_O_4_.H_2_O were used as starting materials for the preparation of copper oxalates, with distilled water and glycerol as solvents.

### 2.2. Synthesis Process of CuO-NPs

As shown in Figure 1, the copper oxalates were prepared by the reflux method carried out at 90 °C. Copper chloride and oxalate ion precursor were separately dissolved in water or water–glycerol mixture according to the data in Table 1. A blue-green precipitate was directly formed after mixing the two solutions. Then, 3 h later, the system was cooled to room temperature before being centrifuged (6000 rpm, 20 min), washed with distilled water, and dried overnight at 50 °C. The resulting dried precipitates were then heated at 400 °C for 4 h with a ramping rate of 5 °C/min in a tube furnace. Black powders were produced and systematically subjected to X-ray powder diffraction analysis to ensure their purity. 

It should be noted that the calcination temperature of 400 °C was selected based on the study of Christensen et al. [24] that demonstrated the formation of pure CuO at temperatures above 345 °C. 

The different codes used to identify the copper oxalate and oxide samples are listed in Table 1.

### 2.3. Characterization Techniques

X-ray powder diffraction (XRPD) analysis of the as-prepared products was performed on a Panalytical X’Pert MPD diffractometer (Malvern, UK) using Cu Kα radiation. The patterns were recorded in the 2θ range between 10° and 100° with a step size of 0.03° and a step time of 3 s. SEM and TEM images were obtained using a JEOL 6400 JSM scanning electron microscope and a JEOL JEM-2100 transmission electron microscope at an accelerating voltage of 25 and 200 kV, respectively (Tokyo, Japan). The average particle sizes were determined using the ImageJ-win 32 software program. The magnetic properties were explored with a Quantum Design SQUID MPSM-XL magnetometer (Quantum Design, GmbH, Darmstadt, Germany). The hysteresis curves (M-H) were measured at 5 and 300 K between −5 and +5 T. Zero-field-cooled and field-cooled curves (ZFC-FC) were recorded from 5 to 300 K at an applied field of 1 kOe. 

### 2.4. Theoretical Background

#### 2.4.1. Estimation of the Average Crystallite Size

The average crystallite sizes (D) for CuO powders were calculated using the following Debye–Scherrer equation where K is the Scherrer constant (K = 0.9) related to the crystallite shape, λ (nm) and θ (rad) are the X-ray radiation wavelength and the Bragg’s angle, respectively, and β (rad) is the full width at half maximum (FWHM) of the diffraction peaks determined by a pseudo-Voigt profile fit [25]: (1)$$D=\frac{K\lambda }{\beta \cos\theta }$$

#### 2.4.2. Hysteresis Loops Susceptibility Analysis

At low fields (LF), the susceptibility is the sum of all magnetic contributions, i.e., paramagnetic (PM), diamagnetic (DM), antiferromagnetic (AFM), and ferromagnetic-like (FM), and can be described as follows (Equation (2)) [26]: (2)$$\chi_{\mathrm{LF}}=\chi_{\mathrm{PM}}+\chi_{\mathrm{DM}}+\chi_{\mathrm{AFM}}+\chi_{\mathrm{FM}}$$

On the other hand, because the FM contributions saturate at high fields (HF), only PM, DM, and AFM ordering contribute to the susceptibility. Therefore, FM-like contributions can be discarded at HF, and high field susceptibility can be described using Equation (3) as:(3)$$\chi_{\mathrm{HF}}=\chi_{\mathrm{PM}}+\chi_{\mathrm{DM}}+\chi_{\mathrm{AFM}}$$

Consequently, FM contributions can be evaluated by calculating the difference represented by Equation (4):(4)$$\chi_{\mathrm{FM}}=\chi_{\mathrm{LF}}-\chi_{\mathrm{HF}}$$

Using the previous equations, the hysteresis loops that contain more than one magnetic response can be separately analyzed to study the different magnetic contributions.

## 3. Results and Discussion

### 3.1. X-ray Structural Analysis and TEM Observations

Figure 2 displays the XRPD patterns for the copper oxalates and oxides, which are in agreement with the JCPDS cards no. 21-0297 and 48-1548, respectively. The well-defined diffraction peaks indicate the good crystallinity of the different samples. However, the intensity of these peaks varied depending on the solvent volume ratio mixture and C_2_O_4_^2−^ starting material. The samples prepared in water exhibited the highest peak intensity and so the best crystallinity, whereas an increase in the glycerol amount resulted in a decrease in the peak intensity without losing the crystallinity of the material. This behavior was observed for both the N and H samples. Furthermore, the diffraction peaks for the N samples appeared slightly wider than those for the H samples, indicating the formation of smaller crystallites. This conclusion was quantitatively supported by calculating the average crystallite size D of CuO-NPs using Equation (1) for the two highest peaks (−111) and (111) that were fitted by the pseudo-Voigt function. The obtained results are collected in Table 2. As shown, the smallest crystallites corresponded to the N3 sample with D = 19 nm.

As far as the CuO purity is concerned, no additional peaks were observed for any samples except for N1, N2, and N3 that exhibited a small diffraction peak at 2θ = 36.54°. This latter is a signature of the Cu_2_O phase which crystallizes in the cubic system, Pn3m space group (Figure 2b) [27]. A two-phase (CuO-Cu_2_O) Rietveld refinement was conducted and revealed that N1, N2, and N3 contained 2.29, 5.41, and 5.32% of the Cu_2_O phase, respectively. The final Rietveld plots and the refined cell parameters for all samples are gathered in Figure 3 and Table 2, respectively.

To obtain further insight into the presence of Cu_2_O in N1, N2, and N3 samples, which may be attributed to the incomplete oxidation of Cu_2_O to CuO at 400 °C, N1oxa, N2oxa, and N3oxa were calcined at 500 °C and the resulting oxides were analyzed by XRPD. Appendix A shows the superposition of powder patterns for N1-2-3 samples synthesized at 400 and 500 °C, with magnification from 2θ = 32° to 44° shown in Figure 4, and reveals that the diffraction peak associated with the Cu_2_O phase disappears at 500 °C thereby confirming the hypothesis of incomplete oxidation of Cu_2_O to CuO at 400 °C. 

To evaluate the CuO particle sizes (PS) and agglomeration state, TEM images were recorded as shown in Figure 5. Oval-shaped CuO nanoparticles with smaller sizes and less pronounced agglomeration were observed with an increase of the glycerol amount. The average particle sizes were deduced from log-normal fitting of size distribution histograms as (42.3, 33.8, 33.3, and 27.4 nm) and (36.4, 28.6, 25.2, and 24.2 nm) for (H0, H1, H2, and H3) and (N0, N1, N2, and N3) samples, respectively. A slight difference in PS was observed for 25 (50) and 50% (75%) glycerol amounts in H (N) samples, although a significant decrease in particle size was noticed from the aqueous medium to higher glycerol volumes for both sample types (from 42.3 to 27.4 nm for H0 and H3 and from 36.4 to 24.2 nm for N0 and N3, with a ratio of H0/H3 to N0/N3 of approximately 1.5).

Figure 6 shows a graphical comparison between the crystallite and particle sizes for H and N samples. The difference between PS and D (red and green arrows) is reduced for the samples synthesized in the richest glycerol medium, thus revealing lower agglomeration behavior. 

### 3.2. Scanning Electron Microscopy 

To understand better the effect of using different C_2_O_4_^2−^ ion precursors and volume ratios of water/glycerol mixture solvent while keeping the same reaction temperature, time, and total volume of solvents, a deeper investigation was performed using the SEM images of the copper oxalate and oxide samples, as shown in Figure 7. 

The oxalate samples (Figure 7a) demonstrated a gradual change in shape from quasi-spherical particles for H0oxa, H1oxa, H2oxa, N0oxa, N1oxa, and N2oxa to rice-husk-like and cluster forms for H3oxa and N3oxa samples, respectively, as the glycerol level increased. In addition, some torus structures were also noticed in sample H1oxa, as highlighted by the red circle. 

The morphologies of the copper oxides (Figure 7b) were found to be similar to their corresponding copper oxalate precursors, except for H3oxa and N3oxa prepared with the higher volume of glycerol which showed an evolution from a flat surface and large clusters, respectively, to small clusters with a porous structure. This indicates that the water/glycerol mixed solvent not only reduced the particles size but also led to different surface textures. From the viewpoint of nucleation and growth mechanisms during the synthesis process, it is important to note that cloudy solutions were obtained once the oxalate precursor was added to the copper solution. This suggests a rapid nucleation step and strong interactions between the system molecules. In the aqueous solution, the high ionic mobility led to instantaneous precipitation and spontaneous aggregation of the copper oxalates. However, increasing the glycerol volume made the liquid medium more viscous, as glycerol has a viscosity approximately 1000 times greater than that of water [28]. As a consequence, the ions’ diffusion length was slowed and therefore growth of the particles was prevented. 

According to Zelent et al. [29], different hydrogen bonding interactions can occur between solvent molecules, namely water–water, water–glycerol and glycerol–glycerol. In a pure water solvent, particles are loosely bonded together. As the amount of glycerol increases, the solvent polarity and the H-bonding interactions increase, resulting in the formation of smaller particles. These results agreed with those obtained by Wang et al. for the synthesis of ZnO nanoparticles in water/glycerol solvent [30]. With regard to the effect of the oxalate ions source, the observed differences between Hoxa and Noxa samples were due to the larger size of (NH_4_)^+^ compared to H^+^ which led to greater separation between particles, less agglomeration, and therefore better dispersion. To the best of our knowledge, few studies have reported on oxalic acid and ammonium oxalate as starting materials for the preparation of metal oxalates in terms of their effects on the characteristics of the resulting powders. In particular, Baco-Carles et al. [31] and Nagirnyak et al. [32] respectively studied the correlations between the morphologies of β-CoC_2_O_4_.2H_2_O and SnC_2_O_4_ prepared using H_2_C_2_O_4_.2H_2_O and (NH_4_)_2_C_2_O_4_.H_2_O and the resulting cobalt and tin(IV) oxide powders.

### 3.3. Magnetic Properties of CuO

Figure 8a,b displays the results of the thermal susceptibility measurements (ZFC and FC curves) performed for all the CuO samples in the temperature range 5–300 K under an applied magnetic field of 1000 Oe. As can be seen, independently of the synthesis conditions, the magnetic behavior resembles that of the bulk material at temperatures above 100 K [12]. The calculated susceptibilities varied from 3.3 × 10^−6^ to 3.5 × 10^−6^ emu.g^−1^.Oe^−1^, values very close to those reported by Kobler and Chattopadhyay [33] for CuO single crystals. Punoose et al. [15] also made a similar observation that the susceptibility of particles with a size of 37 nm exhibited comparable behavior to the bulk. 

However, below 100 K, all the ZFC-FC curves split and rose with the appearance of a cusp around 50 K that was not observed in the bulk but has recently been observed in 11 nm CuO nanoparticles [34]. To understand better the thermal susceptibility behavior, the derivative (dχ/dT) of the ZFC curves was calculated from 5 to 300 K, as shown in Figure 8c,d for samples H2 and N2. Two main points were identified: (i) T_g_~50 K where the derivative is null and (ii) T_I_ = 210 K where the derivative reaches a maximum indicating a concavity change in the susceptibility curve (although not shown, all the samples exhibited similar T_g_ and T_I_ values).

The temperature T_I_ = 210 K is close to the incommensurate-to-commensurate antiferromagnetic transition at 213 K reported by Ota and Gmelin [14]. Therefore, our results show that the whole magnetic behavior of the synthesized nanoparticles was close to that of the bulk, but this transition temperature was slightly decreased because of the smaller particle size. 

On the other hand, T_g_ has recently been observed by other authors at 20 K [34]. These cusps can be assigned to a spin glass behavior caused by frustrated spins at the particle surface which coexist with antiferromagnetic interactions inside the particles, and these cusps can move from low to high temperatures as frustration increases [35]. In this work, the particle sizes were around 20–40 nm which can give place to frustration. In addition, the incomplete bindings of the atoms at the surface and the high surface/volume ratio can enhance this frustration [36]. Therefore, the maximum found at T_g_ = 50 K is attributed to competing ferromagnetic—antiferromagnetic interactions inside the nanoparticles, where the ferromagnetic contribution comes from the spin glass frustration. Indeed, the split and rise of the FC and ZFC magnetization curves at temperatures smaller than 100 K confirm the presence of competing antiferromagnetic–ferromagnetic interactions.

For a better understanding of the magnetic behavior of the synthesized CuO-NPs, the hysteresis cycles were measured at 5 and 300 K for all H and N samples, as shown in Figure 9. It is worth noting that saturation magnetization was not observed at any temperature. However, the magnetization at 300 K was greater than at 5 K, in agreement with the higher susceptibility of the bulk sample at room temperature [15,33]. At low temperatures, hysteresis cycles were observed (insets in Figure 9). 

Previous studies on CuO nanoparticles show ferromagnetic hysteresis cycles at low temperatures [37,38,39]. So, to determine the presence of ferromagnetic interactions, the high field susceptibility was extracted from the hysteresis curves according to Equation (4). Two different behaviors were observed in the H and N samples (Figure 10). For H0 and H1, it was not possible to determine a ferromagnetic curve. However, ferromagnetic interactions were detected for H2 and H3 and the hysteresis curve of H3 shows higher magnetization and coercivity than H2 (Table 3). This is quite consistent with the decreasing particle size from H0 → H3, and also supports the results of the thermal magnetization: as the particle size decreases there appear competing ferromagnetic–antiferromagnetic interactions inside the particles due to the contribution of the spin glass at the surface and the still antiferromagnetic core. It is worth noting that these two magnetic phases are coupled, and the hysteresis curves do not show independent contributions of the two phases. This is also supported by the higher coercive field of H3, suggesting that the exchange interactions became stronger with decreasing particle size.

Regarding N samples, the saturation magnetizations as well as the coercive fields remained almost constant for all the samples, consistent with their smaller particle sizes (Table 3). Batsaikhan et al. [40] reported that the onset of FM interaction in CuO strongly depends on the particle size; it seems to reach its maximum at around 10 nm, and then decreases for larger and smaller sizes. The particle sizes in this work were above this limit, and the magnetic behavior resembled that of the bulk, with the exception of a spin glass frustration at the particle surface. Considering that the FM contribution arises from the atoms at the surface, it is possible to estimate the contribution from the surface volume (which is FM) to the core volume (which is still AFM). To estimate the surface volume, a shell of 4.34 Å was considered (which is a pseudocubic cell with the same volume as the CuO cell). For 10 nm NPs, the surface volume is 13% of the whole volume, whereas for 30 nm NPs, the surface volume is smaller than 4%. This result explains the great differences observed in NPs with sizes around 10 nm [34,40] and the magnetization values obtained in this work. 

## 4. Conclusions

In summary, CuO-NPs were synthesized via thermal decomposition of pre-prepared copper oxalates through a reflux process. X-ray powder diffraction analysis was used to determine the purity and the average crystallite size. The analysis revealed the presence of a small amount of Cu_2_O for the samples prepared from oxalate ammonium as the oxalate ion precursor and water/glycerol mixture as the solvent at 25%, 50% and 75% *V/V* ratios. The TEM observations showed similar shapes but different average particle sizes in the range 24–42 nm, according to C_2_O_4_^2−^ starting material and solvent type. The effects of these two parameters on the morphological features of CuO were studied by SEM analysis, which indicated an evolution from a spherical to rice-husk-like particles. The magnetic properties of the CuO-NPs were evaluated using M-T and M-H curves and interpreted by the coexistence of dominant antiferromagnetic and weak ferromagnetic interactions. The ZFC-FC curves showed that the magnetic behavior of these NPs was similar to that of the bulk above 100 K; however, a cusp appeared around 50 K which was attributed to spin glass due to the high frustration existing at the surface of small particles. Ferromagnetic contributions from spin glass at the NPs’ surface were extracted from the hysteresis cycles. In addition, the derivatives of the ZFC curves show that a maximum is reached at 210 K, indicating a concavity change. This temperature is possibly related to the Néel temperature in bulk (230 K), although it is decreased because of the smaller size of the NPs. 

## Figures and Tables

**Figure 1 materials-16-03426-f001:**
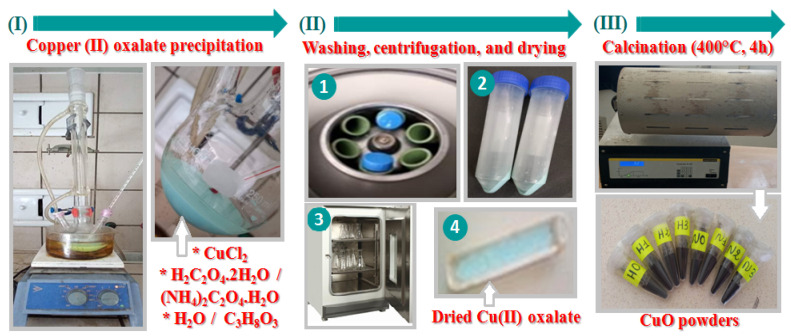
Illustration of the synthesis process of CuO-NPs.

**Figure 2 materials-16-03426-f002:**
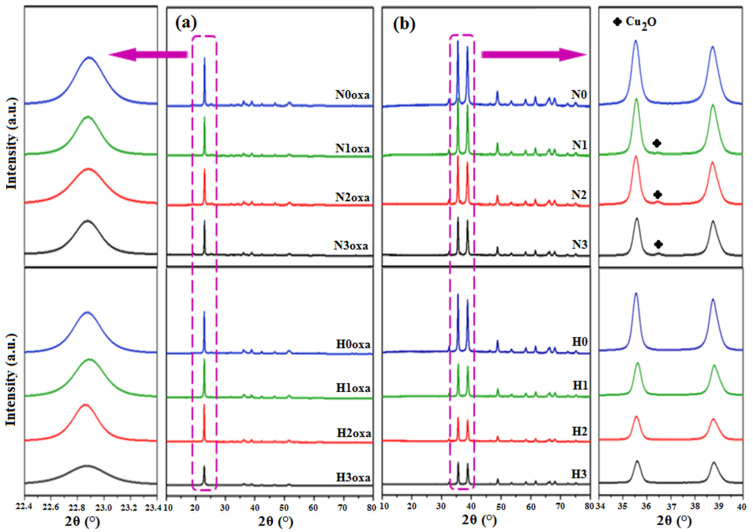
XRPD patterns with magnification of the highest intensity peaks of (**a**) copper oxalate and (**b**) copper oxide powders.

**Figure 3 materials-16-03426-f003:**
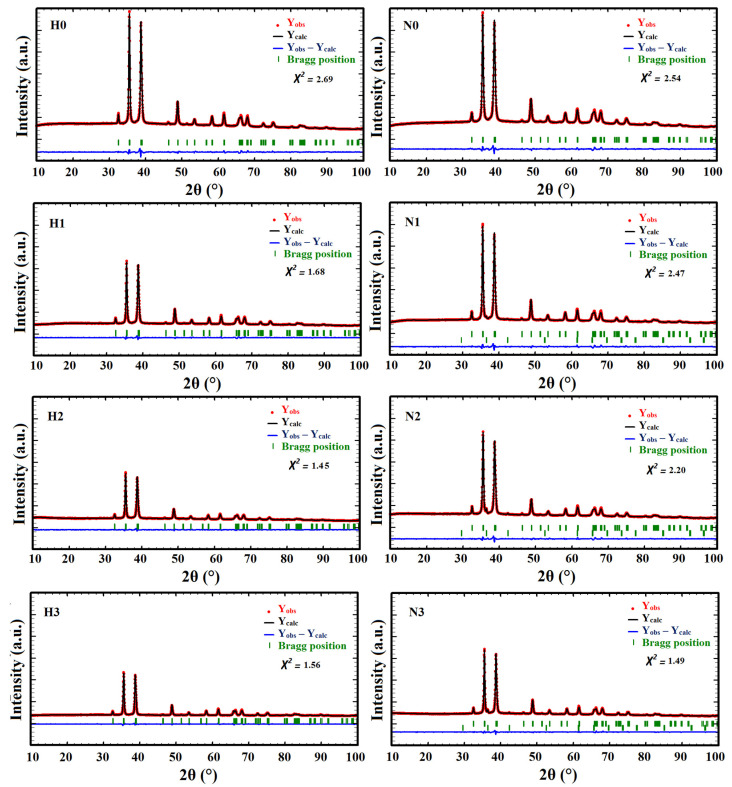
Final Rietveld refinement plots for CuO powder patterns for samples H0–H3 and N0–N3.

**Figure 4 materials-16-03426-f004:**
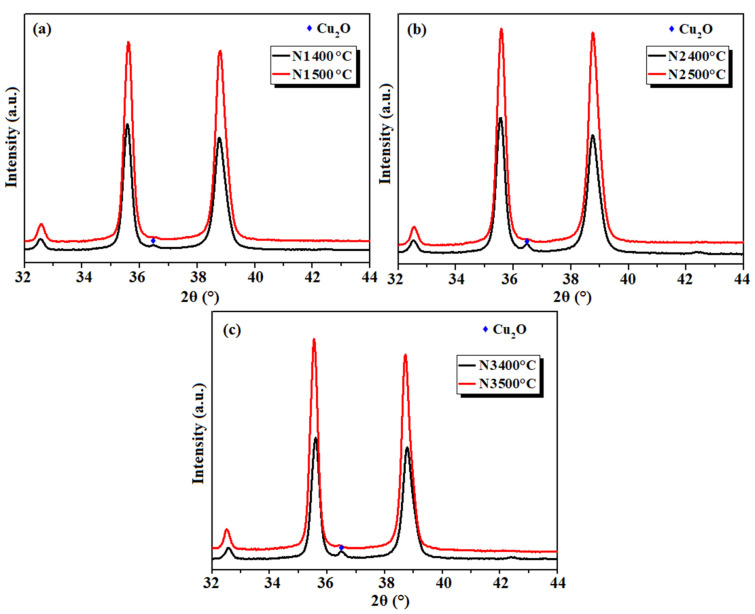
Magnified XRPD patterns (2θ = 32–44°) for (**a**) N1, (**b**) N2, and (**c**) N3 samples synthesized at 400 and 500 °C.

**Figure 5 materials-16-03426-f005:**
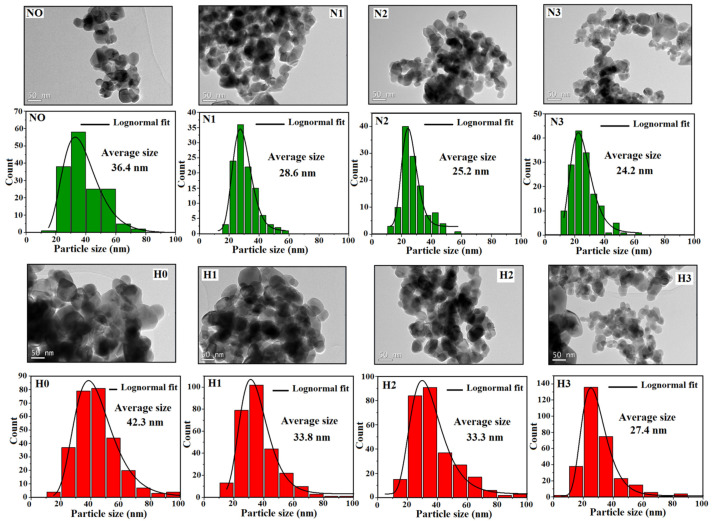
TEM images and size distributions of CuO-NPs.

**Figure 6 materials-16-03426-f006:**
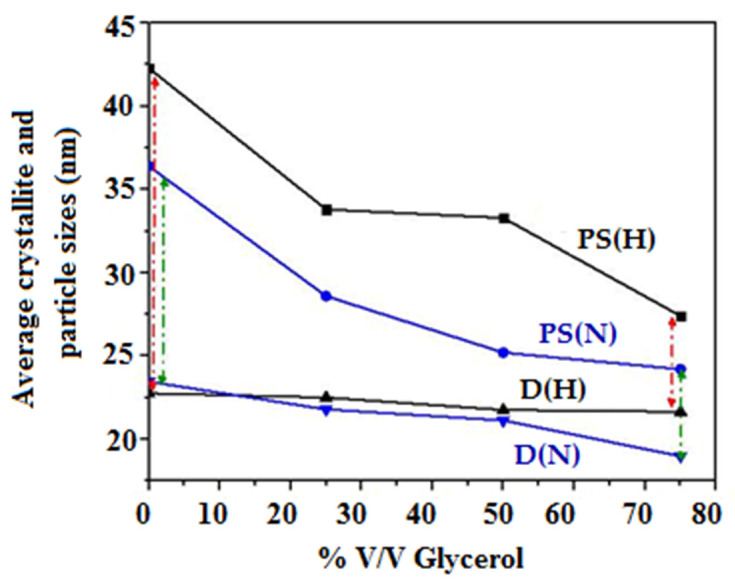
Effect of glycerol amount (% *V*/*V*) on crystallite and particle sizes of H and N samples (red and green arrows show the difference between PS and D for H and N samples, respectively).

**Figure 7 materials-16-03426-f007:**
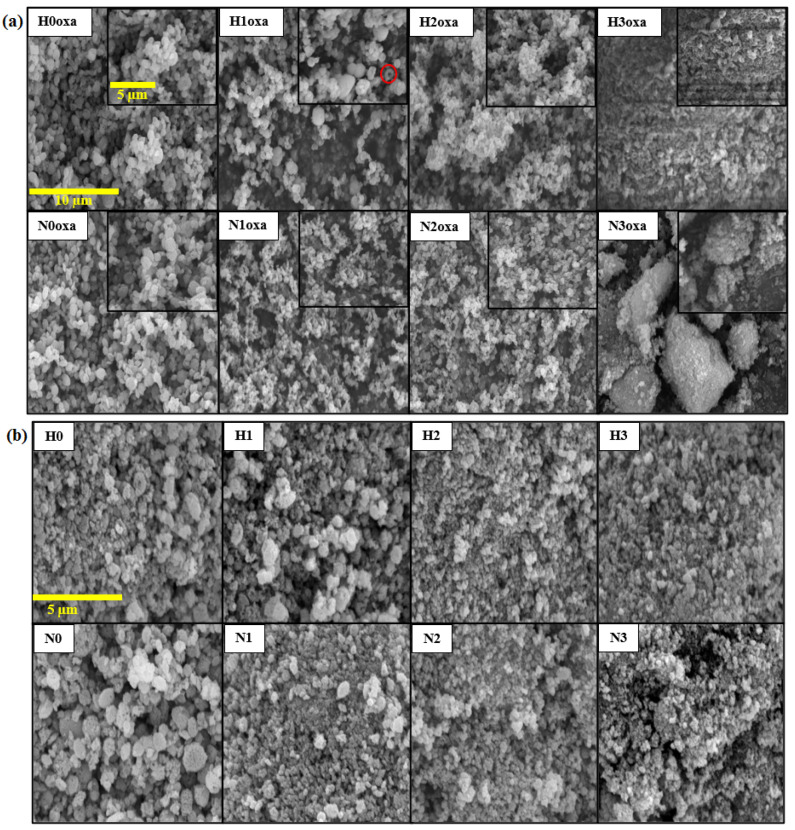
SEM images of (**a**) copper oxalate precursors with two magnifications (The red circle highlights the torus structure observed for H1oxa sample) (**b**) copper oxide powders.

**Figure 8 materials-16-03426-f008:**
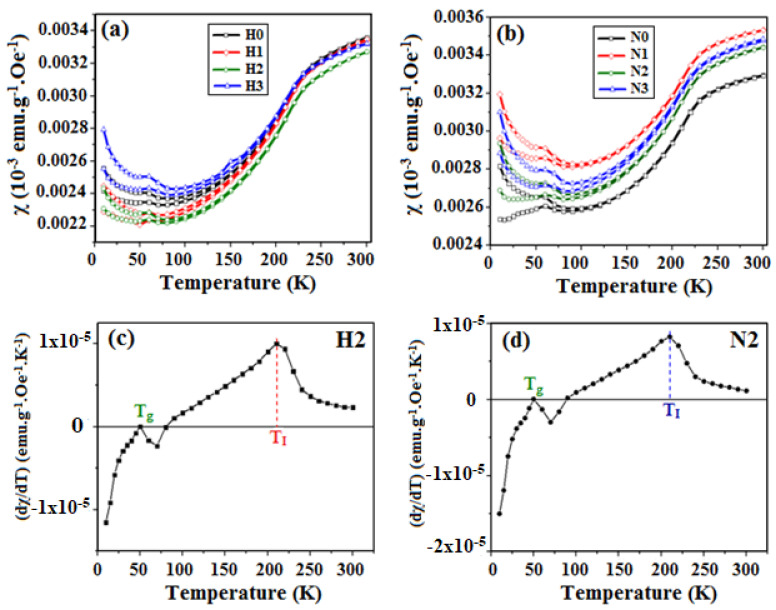
(**a**,**b**) Susceptibility χ obtained from ZFC and FC magnetization curves for H and N samples, respectively. (**c**,**d**) Derivative of the χ curves for samples H2 and N2, respectively.

**Figure 9 materials-16-03426-f009:**
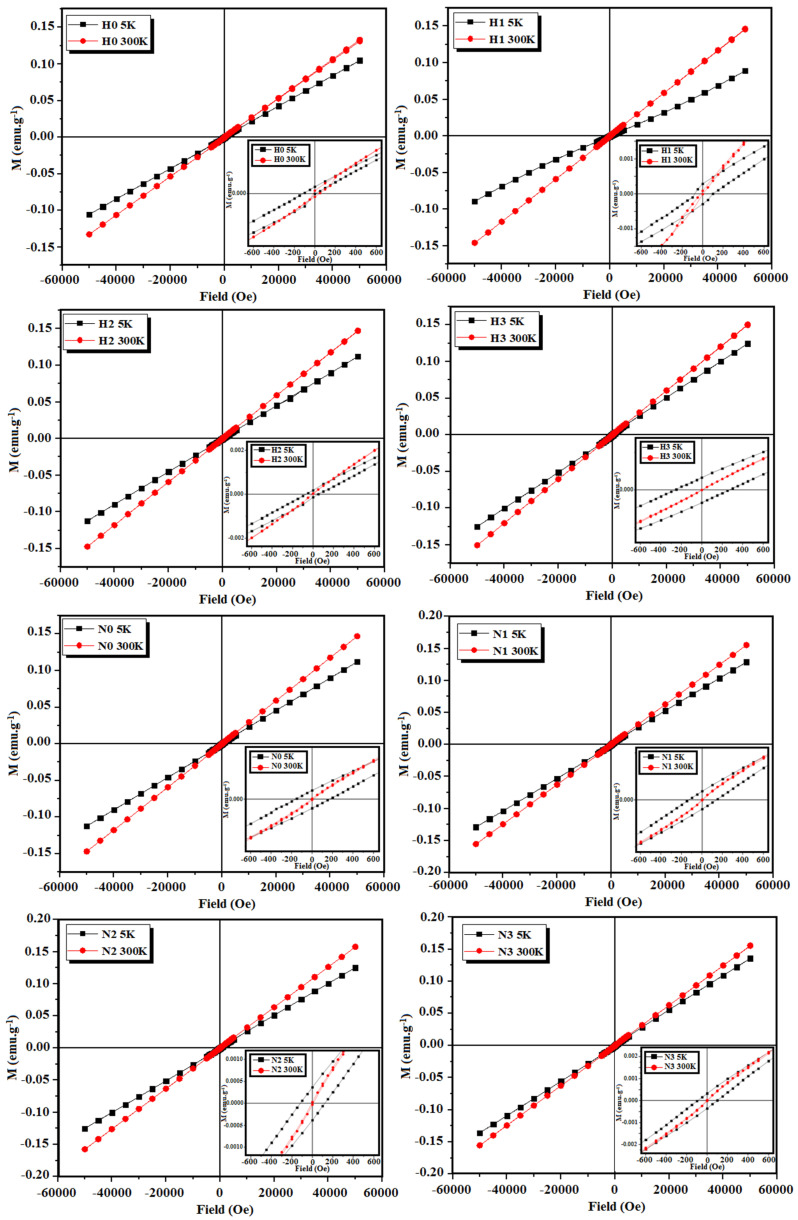
Hysteresis cycles measured at 5 and 300 K for H and N samples.

**Figure 10 materials-16-03426-f010:**
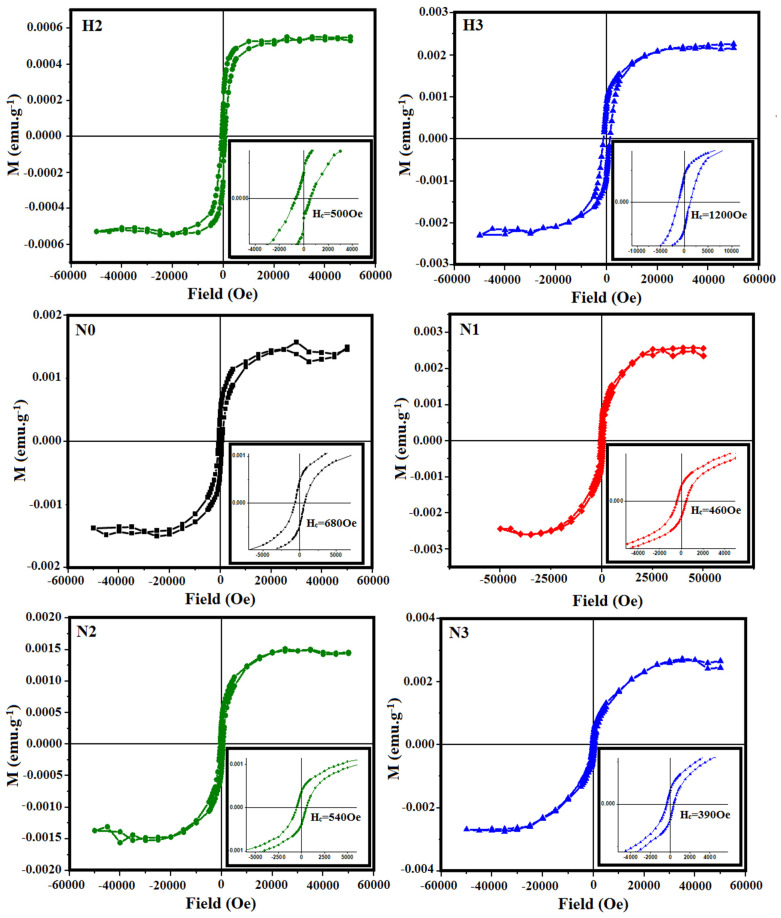
Ferromagnetic cycles extracted from the whole magnetization (Figure 9) using Equation (4) for H (H2, H3) and N samples at 5 K.

**Table 1 materials-16-03426-t001:** Samples’ codes and experimental conditions used for the synthesis of copper oxalate precursors.

Oxalate Ion Source	Water:Glycerol Volume (mL)	Copper Oxalate Code	Copper Oxide Code
H_2_C_2_O_4_.2H_2_O	60:0	H0oxa	H0
	45:15	H1oxa	H1
	30:30	H2oxa	H2
	15:45	H3oxa	H3
(NH_4_)_2_C_2_O_4_.H_2_O	60:0	N0oxa	N0
	45:15	N1oxa	N1
	30:30	N2oxa	N2
	15:45	N3oxa	N3

**Table 2 materials-16-03426-t002:** Average crystallite sizes (nm) and refined cell parameters for CuO.

Samples	H0	H1	H2	H3	N0	N1 *	N2 *	N3 *
Average crystallite sizes								
D_(−111)_ (nm)	25.08	25.62	24.49	24.49	26.35	24.86	24.21	21.47
D_(111)_ (nm)	20.41	19.38	19.03	18.82	20.56	18.73	18.05	16.50
D_averge_ (nm)	22.75	22.50	21.76	21.65	23.45	21.79	21.13	18.98
Lattice parameters (C 2/c space group)								
a (Å)	4.6841	4.6849	4.6838	4.6840	4.6835	4.6849	4.6843	4.6853
b (Å)	3.4274	3.4286	3.4282	3.4280	3.4267	3.4277	3.4277	3.4280
c (Å)	5.1293	5.1318	5.1296	5.1304	5.1292	5.1299	5.1292	5.1321
β (°)	99.420	99.397	99.408	99.394	99.413	99.397	99.399	99.389

* Cu_2_O (%): 2.29, 5.41 and 5.32 for N1, N2, and N3 respectively.

**Table 3 materials-16-03426-t003:** Saturation magnetization (Ms) and Coercivity (Hc) of H (H2, H3) and all N samples.

Sample	Saturation Magnetization: Ms (emu.g^−1^)	Coercivity: Hc (Oe)	Particle Size (nm)
H2	0.0005	500	33.3
H3	0.0023	1200	27.4
N0	0.0014	680	36.4
N1	0.0024	460	28.6
N2	0.0014	540	25.2
N3	0.0025	390	24.2

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
