# Peer review of "Insights into the Synthesis Parameters Effects on the Structural, Morphological, and Magnetic Properties of Copper Oxide Nanoparticles"

_materials, 2023, doi:10.3390/ma16093426_

Round 1
Reviewer 1 Report
1. The SAED results of CuO need be provided.
2. The XPS results of CuO need be provided.
3. In my opinion, the Figure 1 is unnecessary. I suggest authors delete the Figure 1.
Author Response
- The SAED results of CuO need be provided.
The technique SAED is interesting when you are working with HRTEM to examine lattice constant or crystal orientation. The lattice constant has been determined by X-Ray Powder Diffraction (XRPD) analysis and the particle sizes were determined by TEM.
- The XPS results of CuO need be provided.
The XPS is necessary for determination of valence states of metals with different valences in the same phase. In our case, the two valence state of Cu form different oxides, CuO and Cu2O, which have been differentiated as different phases by XRPD.
- In my opinion, the Figure 1 is unnecessary. I suggest authors delete the Figure 1.
Figure 1 has been removed.
Reviewer 2 Report
Copper oxide (CuO) NPs were synthesized using two different oxalic precursors such as oxalic acid, and ammonium oxalate together in water and water plus glycerol mixture as solvent. The resulting species is calcinated at 400-500 oC in an air atmosphere, resulting in the decomposition of copper oxalates into CuO NPs. The resulting CuO NPs are agglomerated but not individual monodisperse particles. The author further investigated the magnetic properties of these particles’ states that coexistence of dominant antiferromagnetic and weak ferromagnetic behavior.
Comments
1) The authors claim that there is a varied size distribution of CuO NPs obtained by controlling the water-to-glycerol ratio and precursor variation by using TEM images. The authors did not state the rationality behind the calcination in air atmosphere but not in O2 to obtain CuO NPs. I believe that the oxygen environment is a more strict condition to obtain the clean NPs such as no variation in the oxygen content in the CuO NPs.
2) I do not understand why there is different magnetization saturation for the different synthesized particles within the small range of different-sized particles. Perhaps, it may be visual to plot the tabulated data and see any relation with precursor concentration, size or any other parameter which fits with the saturation could help to improve the understanding.
Author Response
THe answers are in the attached document.
